# Muscleblind-like proteins are novel modulators of the tumor-immune microenvironment

**Austin M. Gabel**[1,2,3,4☺], **Edie I. Crosse**[1,2☺], **Andrea E. Belleville**[1,2,4,5], **Simon J. Hogg**[6], **Siegen A. McKellar**[1,2,4,5], **Omar Abdel-Wahab**[6‡], **James D. Thomas**[1,2‡], **Robert K. Bradley**[1,2,3‡]*

1 Computational Biology Program, Public Health Sciences Division, Fred Hutchinson Cancer Center, Seattle, Washington, United States of America, 2 Basic Sciences Division, Fred Hutchinson Cancer Center, Seattle, Washington, United States of America, 3 Department of Genome Sciences, University of Washington, Seattle, Washington, United States of America, 4 Medical Scientist Training Program, University of Washington, Seattle, Washington, United States of America, 5 Molecular and Cellular Biology Program, University of Washington, Seattle, Washington, United States of America, 6 Molecular Pharmacology Program, Sloan Kettering Institute, Memorial Sloan Kettering Cancer Center, New York, New York, United States of America

* rbradley@fredhutch.org

☺ These authors contributed equally to this work.

‡ Co-senior authors

## Abstract

Exploiting the immune system to eradicate cancer cells is an area of intense clinical study. However, the mechanisms that shape the tumor-immune microenvironment are incompletely understood. Here, we identify Muscleblind-like (MBNL) proteins as novel modulators of the tumor-immune microenvironment across diverse cancers. We demonstrate that loss of tumor MBNL expression results in an attenuated response to interferon gamma and reduced tumor antigen presentation in melanoma, breast cancer, and colorectal cancer cells. Parallel experiments in a syngeneic mouse melanoma model revealed that MBNL loss reduces tumor cell killing by CD8+ T cells in vitro and facilitates tumor escape from cytotoxic CD8+ T cell infiltration in vivo. Finally, we extended these studies to 29 human cancer types to find that MBNL expression levels are strongly associated with gene expression signatures of T cell tumor infiltration. These insights suggest that MBNL proteins play important roles in shaping the immune landscape across diverse malignancies.

## Introduction

Immunotherapy, which exploits the power of the immune system to fight malignant cells, has transformed cancer treatment within the past decade. Immunotherapies are exemplified by both chimeric antigen receptor (CAR) T cells, designed with high specificity for tumor antigens, and immune checkpoint blockade inhibitors, which block tumor immune evasion mechanisms from cytotoxic T cell killing of cancer cells. The number of different strategies is rapidly growing, including those that recruit natural

**Data availability statement:** The RNA-sequencing data reported in this paper have been deposited in the Gene Expression Omnibus (GEO accession number: GSE261415).

**Funding:** A.M.G and S.A.M are ARCS Foundation scholars. E.I.C is a Damon Runyon fellow. J.D.T. was supported by the NIH/NCI (K99 CA263168). R.K.B. was supported in part by the NIH/NCI (R01 CA251138), NIH/NHLBI (R01 HL128239, R01 HL151651) and the Blood Cancer Discoveries Grant program through the Leukemia & Lymphoma Society, Mark Foundation for Cancer Research, and Paul G. Allen Frontiers Group (8023-20). R.K.B is a Scholar of The Leukemia & Lymphoma Society (1344-18) and holds the McIlwain Family Endowed Chair in Data Science. Computational studies were supported in part by FHCC's Scientific Computing Infrastructure (ORIP S10 OD028685). Experimental studies were supported in part by the Experimental Histopathology, Flow Cytometry, and Genomics Shared Resources of the Fred Hutch/University of Washington Cancer Consortium (NIH/NCI P30 CA015704). The funders had no role in study design, data collection and analysis, decision to publish, or preparation of the manuscript.

**Competing interests:** RKB is a founder and scientific advisor of Codify Therapeutics and Synthesize Bio and holds equity in both companies. OA-W is a founder and scientific advisor of Codify Therapeutics and holds equity in this company. RKB and OA-W have received research funding from Codify Therapeutics unrelated to the current work. OA-W has served as a consultant for Foundation Medicine Inc., Merck, Prelude Therapeutics, Amphista Therapeutics, MagnetBio, and Janssen, and is on the Scientific Advisory Board of Envisagenics Inc., Harmonic Discovery Inc., and Pfizer Boulder; OA-W has received prior research funding from H3B Biomedicine, Nurix Therapeutics, Minovia Therapeutics, and LOXO Oncology unrelated to the current manuscript. This does not alter our adherence to PLoS One policies on sharing data and materials. The remaining authors declare no competing interests.

killer (NK) cells and macrophages, but the currently most clinically advanced therapies are designed to stimulate a robust T cell response directed against the tumor cells [1]. This growing field has demonstrated remarkable successes, such as the complete remission of patients with B cell acute lymphoblastic leukemia following treatment with CAR-T cells targeting CD19 + B cells [2], and immune checkpoint inhibitors such as Ipilimumab (anti-CTLA-4), Pembrolizumab, and Nivolumab (anti-PD-1) have significantly improved survival in patients with metastatic melanoma [3].

Despite these clinical breakthroughs, the efficacy of immunotherapies varies significantly across cancer types and among individual patients. Particularly for immune checkpoint blockade, a crucial determinant for immunotherapeutic success is the extent of T cell immune infiltration within the tumor microenvironment. Indeed, 'hot' tumors, characterized by substantial adaptive immune responses, have been linked with more effective responses to checkpoint blockade [4]. Thus, deciphering the mechanisms that shape the tumor-immune microenvironment is essential for devising strategies to transform 'cold' tumors, those with minimal immune infiltration, into 'hot,' immunoreactive ones.

In this context, our report unveils a role for Muscleblind-like (MBNL) proteins as novel regulators of resistance to T cell-mediated tumor killing through alterations in antigen presentation. In the human genome, the three MBNL genes encode RNA-binding proteins with roles including regulation of alternative RNA splicing, polyadenylation, and localization [5–7]. The pathogenesis of MBNL dysregulation is best described for myotonic dystrophy [8], a neuromuscular disease caused by loss of MBNL function that results in myotonia, cardiomyopathy, and central nervous system abnormalities. Additionally, there is growing evidence for MBNL's role in cancer. MBNL expression is downregulated in colorectal [9] and prostate [10] cancers. In the case of prostate cancer, however, there is additionally increased inclusion of MBNL exon 7, associated with pro-tumor activity and indicative of a splice isoform-specific role in cancer. Conversely, MBNL1 is consistently overexpressed in MLL-rearranged leukemia [11]. Furthermore, MBNL1 has been reported to play roles in the inhibition of glioblastoma tumor initiation [12] and suppression of metastasis in breast [13] and colon [14] cancers.

While most studies have focused on cancer cell-intrinsic roles of MBNLs, little is known about how they may shape the tumor microenvironment. In this study, we investigated and characterized a novel role for MBNL proteins in anti-tumor immune responses. Using mouse models of melanoma, we determine that MBNL loss attenuates MHC Class I antigen presentation, response to immune stimulation by interferon gamma, and subsequent cancer cell killing by cytotoxic T cells. Furthermore, we demonstrate that our results from mouse melanoma extend to multiple additional mouse cancer models and are consistent with transcriptional signatures of tumor-immune interactions from diverse clinical human cancers. The correlation between MBNL and T cell infiltration that we initially identified in mouse cancer models extends across many human cancer types based on genomic analyses of human clinical datasets. Overall, our study highlights an important role for MBNL proteins in shaping the tumor microenvironment, broadening our understanding of the intricacies of tumor-immune interactions.

## Results

### MBNL loss confers resistance to cytotoxic T cell killing of mouse melanoma cells

In a prior publication by Pan et al. [15], a genome-wide CRISPR-Cas9 screen was carried out on B16-F10 mouse melanoma cells to identify genes whose loss conferred resistance to antigen-specific killing by cytotoxic T cells. We mined this dataset to identify novel modulators of anti-tumor immune responses that were not highlighted or further validated in Pan et al.'s study. The screen first involved a pre-treatment with or without 1 ng/mL of IFNγ for 24 hours to stimulate MHC Class I surface expression, as B16-F10 cells have inherently low cell-surface MHC Class I levels. Then, B16-F10 cells were co-cultured with either of two types of cytotoxic T cells: (i) Pmel-1, which recognizes the endogenous melanoma antigen gp100, or (ii) OT-1, which, while having a low affinity for gp100, strongly recognizes Ova peptide (OVA), which was provided exogenously to tumor cells.

As expected and reported by Pan et al, investigation of the ranked list of genes enriched and depleted in the Pmel-1 screen by log fold change identified *B2m* as a top hit conferring resistance to T cell-mediated tumor killing (Fig 1A). B2M is a component of the MHC Class I complex and thus critical for T cell receptor recognition. We were intrigued to find that two members of the Muscleblind-like protein family, MBNL1 and MBNL2, also ranked very highly, indicative of a previously undescribed role for MBNL proteins in mediating cellular response to cytotoxic T cell killing.

A focused examination of *B2m*, *Mbnl1*, and *Mbnl2* demonstrated a significant enrichment for gRNAs targeting all three of these genes when in the presence of Pmel-1 CD8+ cells (Fig 1B) following treatment with IFNγ. Of the two *Mbnl* genes enriched in the screen, *Mbnl1* exhibited the larger positive log fold-change relative to both tumor cells without T cells and those co-cultured with irrelevant T cells that do not recognize the tumor antigen (Fig 1A, B). The same trend was seen for co-culture with OT-I T cells when stimulated with the exogenous antigen, OVA peptide (Fig 1C). Taken together, these data indicate that Mbnl1 and, to a lesser degree, Mbnl2, have direct roles in modulating MHC Class I-dependent killing of cancer cells by antigen-specific T cells.

### MBNL modulates antigen presentation in response to IFNγ

Motivated by our re-analysis of the data from Pan et al., we next sought to further elucidate the role of MBNL proteins in shaping the tumor-immune microenvironment. We generated CRISPR-Cas9 knockout models of *Mbnl1* ("1KO") and *Mbnl2* ("2KO") in B16-F10 melanoma cells (Fig 2A). Although western blotting of polyclonal cells did not reveal complete absence of protein, we confirmed that the majority of alleles in the polyclonal population harbored indels via next-generation DNA sequencing (Fig S1A–S1D in S1 File). Additionally, we generated a double knockout (DKO) of both *Mbnl1* and *Mbnl2*, as MBNL1 and MBNL2 are partially functionally redundant, and there is evidence that MBNL2 can be upregulated to compensate for MBNL1 loss [16]. Indeed, we observed an increase in MBNL2 protein levels with *Mbnl1* KO (Fig 2A), emphasizing the importance of an *Mbnl* DKO model system to assess the role of MBNL proteins in IFNγ signaling.

To assess transcriptional alterations in response to IFNγ, we performed RNA-seq of *Mbnl* DKO cells compared to B16-F10 cells treated with a non-targeting control (NTC) gRNA treated with or without 1 ng/mL IFNγ for 24 hours. We first validated our IFNγ treatment by comparing treated and untreated cells and found that IFNγ response was the top enriched pathway in treated cells (Fig S2A in S1 File; S1 Table). As expected, top-ranking genes contributing to the pathway's enrichment are involved in antigen presentation (Fig 2B). This includes proteasomes involved in antigen processing for presentation (*Psmb2*, *Psmb8*, *Psmb9*, *Psmb10*, *Psme1*, *Psme2,* and *Psma3*), *Tap1,* which transports antigens to the ER for attachment to MHC components, and components of the MHC Class I complex itself, including *B2m* and *H2-D1*.

We next used these data to determine how MBNL loss altered the IFNγ response by conducting a comparative analysis of the fold change in these enriched IFNγ response genes between treated and untreated groups for both NTC and *Mbnl* DKO cells. This revealed a substantial attenuation of the interferon response in *Mbnl* DKO cells (Fig 2C), including a diminished IFNγ-dependent induction of genes involved in antigen presentation. Notably, *B2m* expression exhibited significant attenuation in response to IFNγ for DKO cells (Fig 2D), suggesting that MBNL proteins contribute to the modulation

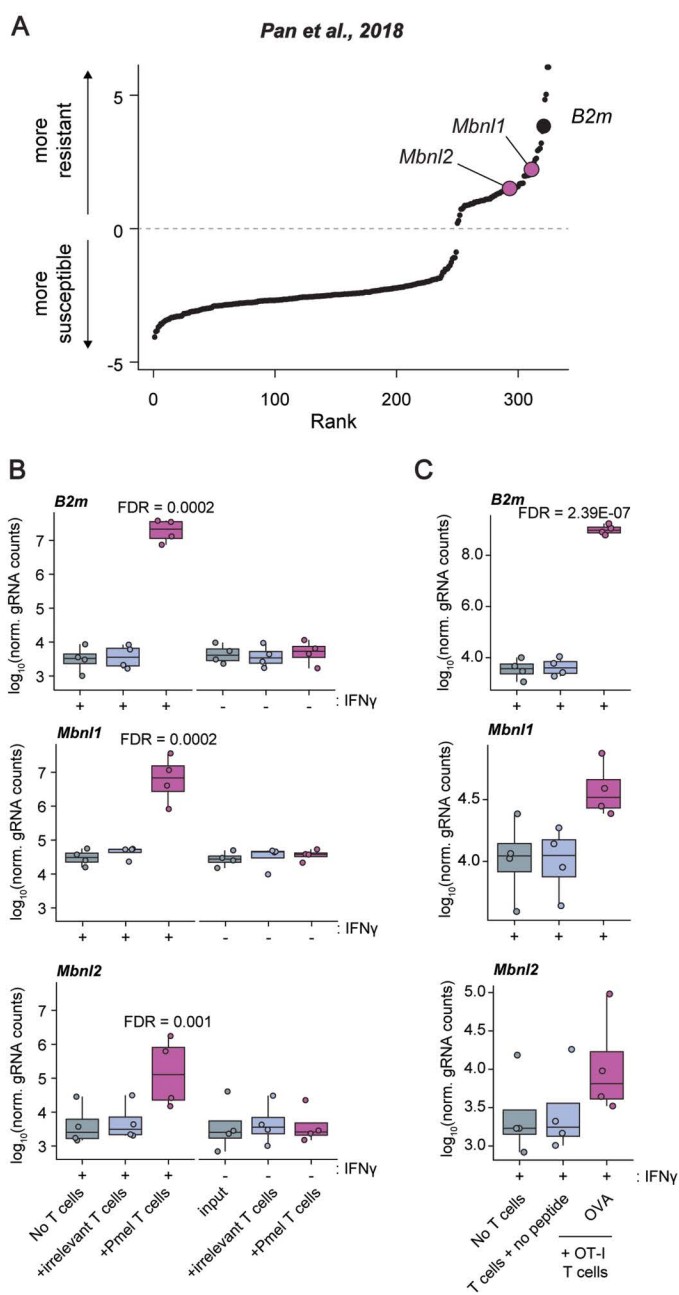

**Fig 1. MBNL loss confers resistance to cytotoxic T cell killing of tumor cells.** (**A**) Rank plot of log fold-change of gRNA enrichment/depletion from a genome-wide CRISPR-Cas9 screen in B16-F10 melanoma cells under cytotoxic T cell selection pressure (data from Pan et al., 2018). gRNA enrichment indicates a gene whose loss confers resistance to cytotoxic T cell killing. (**B**) Log10 fold-change of normalized counts for *B2m*, *Mbnl1* and *Mbnl2* gRNA knockouts for T cell selection assays with or without irrelevant T cells, Pmel-1 T cells and with or without IFNγ stimulation.(**C**) Log10 fold-change of normalized counts for *B2m*, *Mbnl1* and *Mbnl2* gRNA knockouts for T cell selection assays stimulated with IFNγ with or without OT-1 T cells and with or without exogenous antigen OVA peptide.

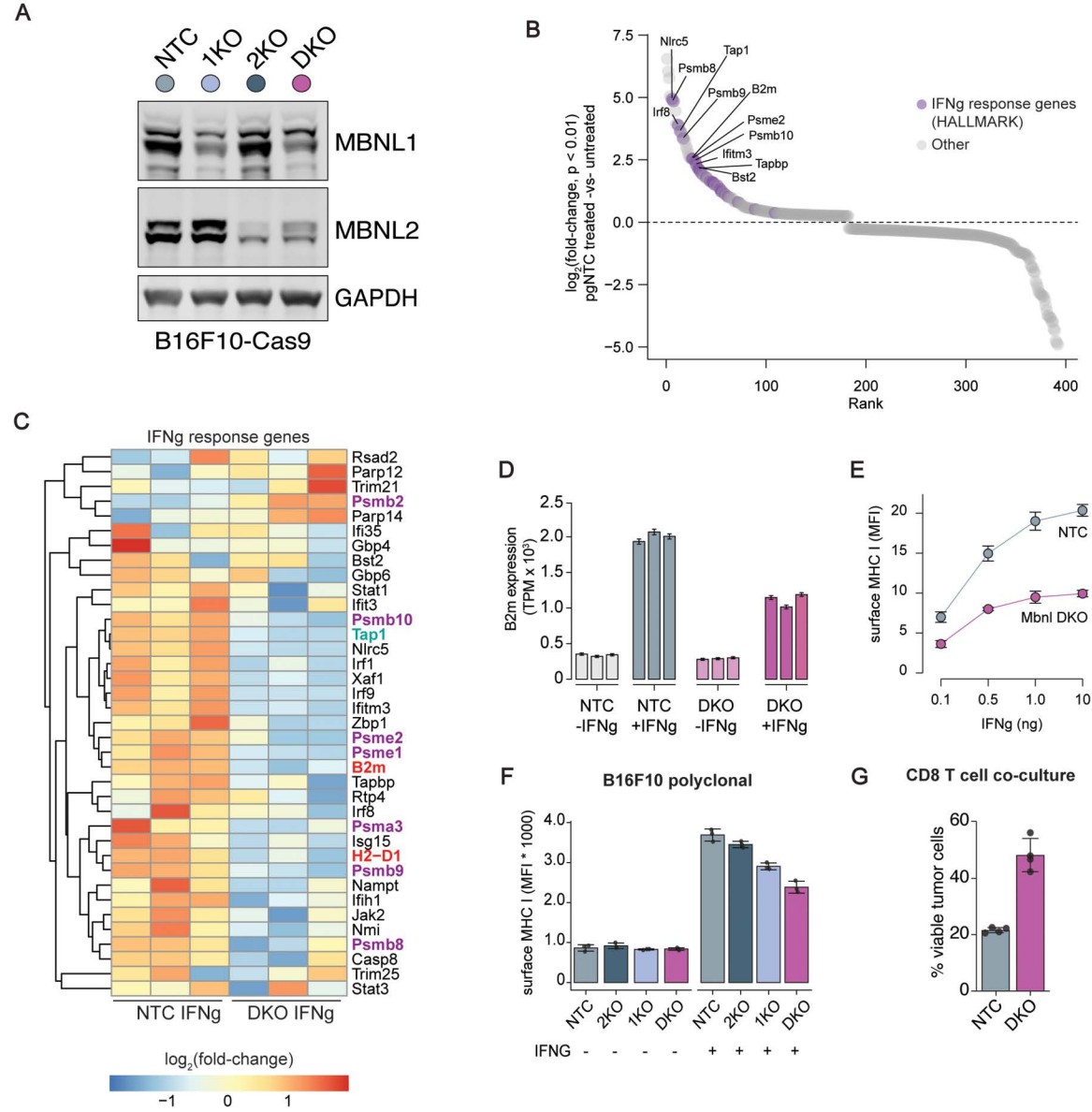

**Fig 2. MBNL modulates antigen presentation in response to IFN γ.** (**A**) Western blot of B16-F10 Cas9 cells treated with the indicated gRNA or pgRNA (NTC = non-targeting control, 1KO = Mbnl1 KO, 2KO = Mbnl2 KO, DKO = Mbnl1 and Mbnl2 double KO). (**B**) Rank plot of log2 fold-change of genes in non-targeting control (NTC) treated vs untreated. Hallmark IFNγ response genes are highlighted in purple and selected genes also involved in MHC antigen presentation are labeled. (**C**) Heatmap of log2 fold-change for either NTC or double knockout (DKO) stimulated with IFNg compared to their unstimulated counterparts. Genes involved in antigen processing, transport to endoplasmic reticulum and components of the major histocompatibility complex are indicated with purple, green and red labels respectively. (**D**) Transcripts per million normalized *B2m* expression for both NTC and DKO with and without IFNγ stimulation. (**E**) Surface MHC Class I (MHC I) expression measured by flow cytometry after stimulation with an IFNγ dilution series. (**F**) Surface MHC I expression with and without 1 ng IFNγ stimulation for NTC, *Mbnl2* knockout (2KO), *Mbnl1* knockout (1KO) and DKO in a B16-F10 polyclonal cell line. (**G**) Percentage viable NTC and DKO B16-F10 tumor cells pulsed with OVA peptide after *in vitro* co-culture with OT-I CD8 + T cells.

of the tumor-immune interaction in part via alterations in expression levels of components of MHC Class I antigen presentation.

We then set out to validate that reduced levels of MHC Class I in MBNL-deficient cells contribute to the reduced T cell cytotoxic activity seen in the Pan et al. screen. We found that *Mbnl* DKO cells had significantly reduced levels of surface MHC Class I compared to NTC cells across all tested IFNγ concentrations, again demonstrating a suppression of induction of MHC Class I components in response to immune stimulation (Fig 2E). The single knockouts also reduced MHC Class I surface expression, with *Mbnl1* KO (1KO) having a larger effect than *Mbnl2* KO (2KO). However, *Mbnl* DKO had the most substantial reduction in MHC Class I, again suggestive of a functional redundancy between *Mbnl1* and *Mbnl2* (Fig 2F). As these experiments were performed in a polyclonal cell line with varying degrees of MBNL protein loss (Fig 2A), we generated genetically validated monoclonal cell lines and repeated the experiments for certainty of the results (Figure S2B-C in S1 File). Once again, surface expression of MHC Class I was significantly reduced ($p$ = 0.0013) in the monoclonal DKO as compared with NTC cells (Fig S2D in S1 File).

We next tested whether a dependence on MBNL for MHC induction following IFNγ stimulation was a common feature of cancer cell lines beyond the B16-F10 model. We generated *Mbnl* single and double knockout lines using CRISPR-Cas9 in cell lines representing two different cancer types, E0771 breast cancer and MC38 colorectal carcinoma lines. For both of these cell lines, we observed a reduction in surface MHC expression for 1KO, 2KO and DKO (Figure S2E-F in S1 File). As expected from the B16-F10 cell data, *Mbnl* DKO reduced surface MHC Class I levels to a greater degree than *Mbnl1* or *Mbnl2* KO alone. Taken together, this provides additional validation and broader relevance for the role of MBNL in regulating tumor antigen presentation across multiple cancer types.

We next sought to test whether the impaired MHC antigen presentation in *Mbnl* DKO B16-F10 cells affected anti-tumor immune activity. Our re-analysis of data from Pan et al strongly suggested that MBNL loss should potentiate immune evasion, but as those results were conducted in the context of a highly multiplexed screen with single gene knockouts, we sought to test how dual knockout of *Mbnl1/2* influenced T cell evasion in a focused assay. We pulsed B16-F10 Cas9 edited cells with OVA peptides and then co-cultured each cell line with OT-I specific CD8 + T cells. We observed a significant increase in the percentage of viable tumor cells after 24 hours in DKO cells compared to NTC cells (Fig 2G). These data indicate that MBNL deficiency and the resultant impaired MHC antigen presentation promotes cancer cell evasion of antigen-specific CD8 + T cell-mediated cytotoxicity.

## MBNL loss facilitates resistance to T cell infiltration in vivo

Having established a role for MBNL in IFNγ response, MHC Class I antigen presentation and T cell-mediated tumor cell killing *in vitro*, we hypothesized that downregulation of MBNL would cause reduced tumor immune infiltration *in vivo.* We transplanted NTC, 1KO, and DKO cells into immunocompetent mice (Fig 3A). We did not observe any significant changes in overall tumor volume (Fig S3A in S1 File); however, we did observe a significant increase in host survival for DKO mice (Figure S3B in S1 File). Given that perturbation of *Mbnl* proteins in disease is known to be deleterious to cell health, we were not surprised to observe a slight reduction in tumor growth in DKO tumors. For example, *Mbnl1* and *Mbnl2* sequestration is known to contribute significantly to cellular dysfunction via RNA misprocessing in neuromuscular disease [17–19]. We screened publicly available data to find that CRISPR knockout of *MBNL1* does not directly affect cancer cell viability (Figure S3C in S1 File). Thus, the combination of *Mbnl1* and *Mbnl2* DKO is likely more deleterious to cancer cell growth than the knockout of either gene alone, again highlighting the likely functional redundancy of these proteins.

Analysis of the dissected tumors showed a reduction in MHC Class I surface expression to levels approaching that of a *B2m* KO control (Fig 3B and 3C). We assessed immune infiltration of the tumors by flow cytometry (Fig S3D in S1 File) and found a reduction in T cells (Fig 3D). Within this T cell population, there was a reduction in CD8 + cells and, to a lesser degree, CD4 + cells (Fig 3D; Figure S3E in S1 File). Similarly, there was a significant reduction in NK cells, potentially indicating a reduced anti-tumor immune activity that is not restricted to T cells (Fig 3D). Correspondingly, we assessed

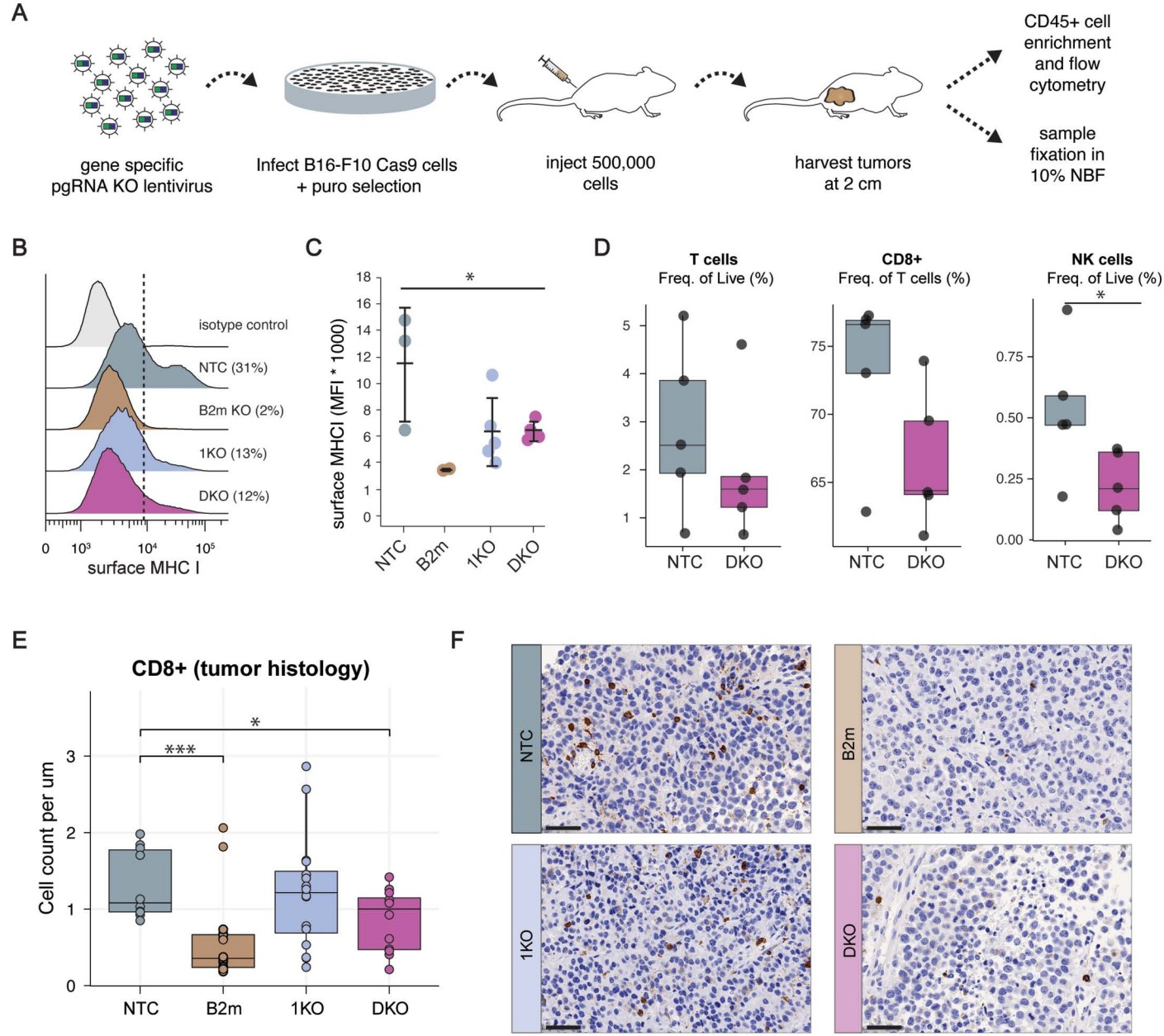

**Fig 3. MBNL loss facilitates resistance to T cell infiltration *in vivo*.** (**A**) Experimental design schematic. (**B-C**) Surface MHC Class I (MHC I) expression measured by flow cytometry across genotypes for B16-F10 tumors; non-targeting control (NTC), *B2m* knockout, *Mbnl1* knockout (1KO) and double knockout (DKO) (* = p < 0.05 by a two-sided Student's *t-t*est). (**D**) Frequency of tumor immune infiltrate populations measured by flow cytometry; T cells, CD8 + frequency within T cells and NK cells (* = p < 0.05 by a two-sided Student's *t-t*est). (**E**) Quantification of CD8 + cells infiltrates in tumors across genotypes by histology analysis (* = p < 0.05, *** p < 0.0005 by a one-sided Wilcoxon signed-rank test). (**F**) Representative histology images of B16-F10 tumors across genotypes with CD8 + in brown DAB stain.

CD8 + cell frequency in B16-F10 tumors using immunohistochemistry and found a significant reduction in the number of CD8 + cells in DKO B16-F10 tumors relative to NTC B16-F10 tumors (Fig 3E-F). We did not observe any difference in 1KO B16-F10 tumors, which we speculate is due to the corresponding upregulation of *Mbnl2* in *Mbnl1* KO cells (Fig 2A).

Taken together, these data strongly support the hypothesis that MBNL proteins help shape the tumor microenvironment by attenuating anti-tumor immune cell recruitment.

**MBNL1 expression is strongly correlated with anti-tumor immune activity in clinical human cancers**

We next sought to determine whether our findings in mouse tumor models are relevant for human cancers. In the human genome, as in mice, there are three MBNL family genes: *MBNL1, MBNL2,* and *MBNL3*. Using the TCGA melanoma (SKCM) RNA-seq dataset, we correlated *MBNL* expression with *CD8A* gene expression as a proxy for tumor T-cell infiltration (S2 Table). This showed a significant positive correlation (R = 0.27, *p* = 1.8e-08) for *MBNL1*, but not for *MBNL2* or *MBNL3* (Fig 4A).

　　We stratified the SKCM TCGA dataset by low (bottom tercile) and high (top tercile) *MBNL1* expression and quantified changes in gene expression between the two bins. Higher *MBNL1* expression was associated with significantly higher expression of genes related to immune pathways, particularly those relating to T cell activity (Fig 4B). Extending the analysis to other cancer types, we discovered that *MBNL1* expression also had a high positive correlation with *CD8A* expression for several other cancer types, including Testicular Germ Cell Tumors (TCGT) and Pancreatic Ductal Adenocarcinoma (PAAD) (Fig 4A).

　　We correlated *MBNL* expression across all TCGA cancer datasets with tumor T cell infiltration by either *CD8A* expression (Fig 4C–D; S3 Table) or cytolytic T-cell (CYT) score (Fig S4A–B in S1 File). The CYT score is calculated as the geometric mean of the expression of two key cytolytic effectors that are significantly upregulated upon CD8 + T cell activation: granzyme B (*GZMB*) and perforin (*PRF1*) [20]. Strikingly, *MBNL1* showed a significant positive correlation for T cell infiltration (based on *CD8A* expression) for 27 of 29 analyzed TCGA cancer types. To a lesser degree, *MBNL3* had a positive correlation with predicted T cell infiltrations across some cancers, including BRCA and LGG, though it should be noted that *MBNL3* expression is fairly low across all assayed tissues (Fig 4C–D; Figure S4A-B in S1 File Supplementary Figures). Conversely, *MBNL2* trended to have a negative correlation with T cell infiltration (Fig 4C–D; Figure S4A-B in S1 File), which we hypothesize is largely due to the anti-correlated nature of *MBNL1* and *MBNL2* expression given their functional redundancy.

　　We correlated all genes included in the Pan et al. screen CRISPR library with *CD8A* expression across all cancers in the TCGA datasets. *MBNL1* ranked within the top 5% of genes, indicating that *MBNL1* is significantly more positively correlated with *CD8A* expression than are most genes in the genome (Fig 4E). Scoring above it were genes directly involved in interferon response (IFNγ, B2M) and T cell activation (PRF1, GZMA, GZMB, CD274, PDL1). This analysis predicts that MBNL1 has a profound effect on anti-tumor T cell infiltration, similar to that of genes with explicit roles in T cell-mediated killing of cancer cells.

## Discussion

Our study highlights a critical role for MBNL proteins in shaping the tumor-immune microenvironment. In mouse melanoma, this was first revealed through a CRISPR knockout screen, where *Mbnl1* and *Mbnl2* emerged as top hits conferring tumor resistance to cytotoxic T cell-mediated killing [15]. Using our own CRISPR knockout models, we validated this finding by demonstrating reduced antigen-specific CD8+ T cell killing of *Mbnl*-deficient (DKO) cells. Beyond cytotoxic resistance, we observed a dampened transcriptional response to IFNγ stimulation, particularly in genes encoding the antigen presentation machinery and MHC class I complex. *In vivo*, melanoma tumors lacking *Mbnl* exhibited lower MHC class I expression and significantly reduced CD8+ T cell infiltration. Our findings suggest that MBNL loss could contribute to the downregulation of interferon signaling and MHC class I that constitutes a well-documented mechanism of immune evasion found in 40–90% of human tumors [21].

　　Extending our analysis to human cancers, we found that *MBNL1* expression positively correlates with *CD8A* expression and cytolytic T cell activity, suggesting a conserved role for MBNL proteins in regulating T cell infiltration and function.

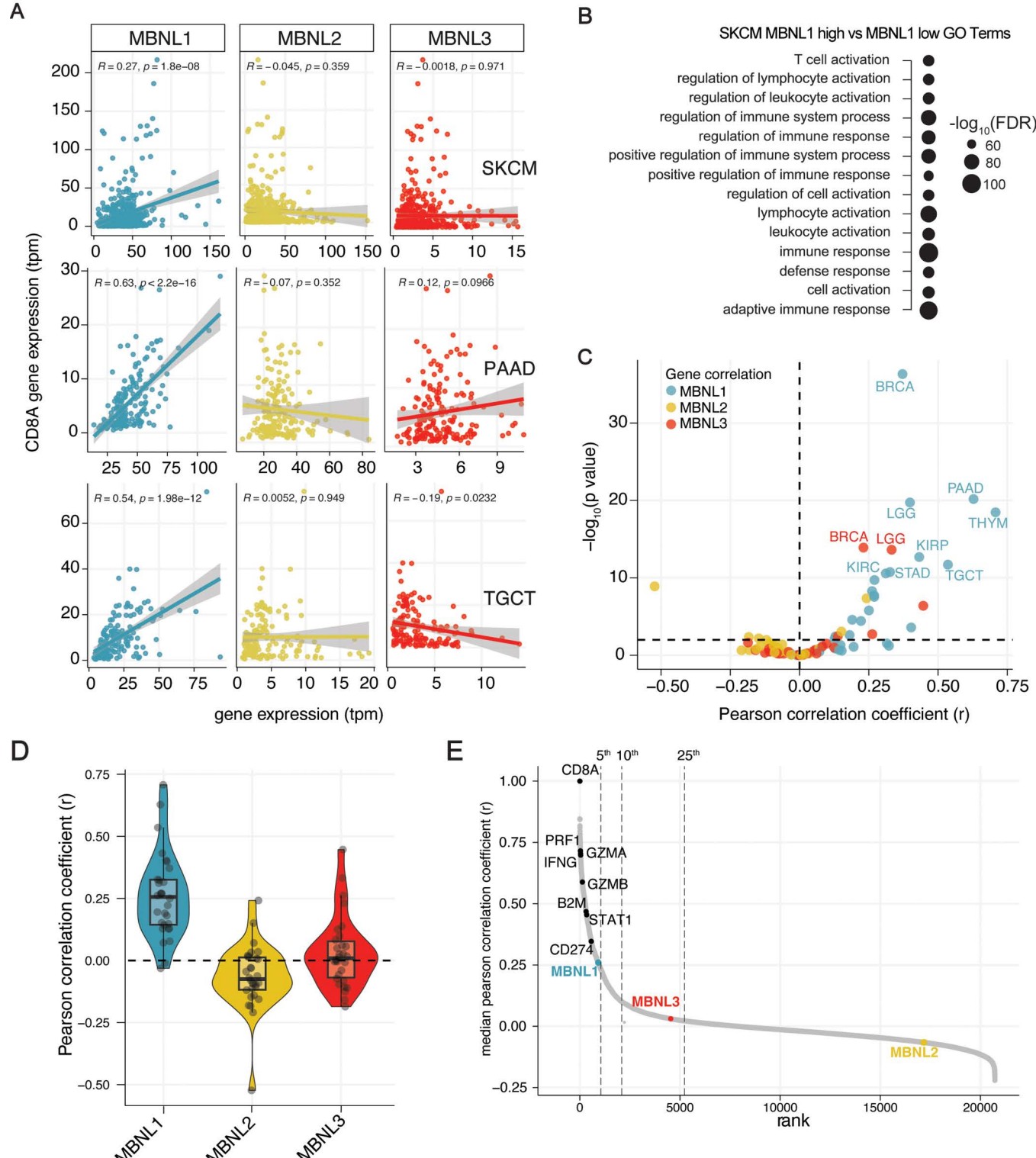

**Fig 4. *MBNL* expression is strongly correlated with anti-tumor immune activity across human cancer types. (A)** Correlation of *MBNL1-3* expression and tumor infiltration indicated by *CD8A* expression (TPM = transcripts per million) for melanoma (SKCM), pancreatic adenocarcinoma (PAAD) and thymoma (THYM). **(B)** Gene Ontology (GO) terms enriched in the upper *MBNL1* expression tercile bottom tercile within the SKCM patient cohort. **(C)**

Plot of Pearson correlation coefficients for MBNL1-3 expression with tumor infiltrate (CD8A expression) across TCGA cancer datasets. (D) Rank plot of the median Pearson correlation coefficient for all gene expression with CD8A expression across all TCGA datasets. MBNL genes and selected immune infiltration related genes are labeled. (E) Waterfall plot of median Pearson correlation coefficient (r) across all analyzed TCGA cohorts comparing the indicated gene expression with gene expression of *CD8A*. Dotted lines indicated the top 5th, 10th and 25th percentiles of positively correlated genes.

Additionally, we present multiple lines of evidence supporting the previously described functional redundancy between MBNL1 and MBNL2 [16, 19]. Collectively, these findings position MBNL proteins as key regulators of the tumor-immune microenvironment.

While our study establishes a link between MBNL expression, antigen presentation, and T cell-mediated tumor killing, the precise molecular mechanisms remain unclear. MBNL proteins are well-characterized modulators of RNA metabolism, influencing mRNA splicing, localization, and stability [5–7, 19]. In several solid tumors, reduced *MBNL1* expression leads to exon skipping in *MAP2K7*, promoting stem cell-like phenotypes and correlating with poor prognosis [22]. Conversely, *MBNL1* overexpression in MLL-rearranged leukemias drives widespread intron removal, altering the expression of key epigenetic modifiers involved in leukemogenesis [11]. It is plausible that MBNL proteins regulate the RNA processing of MHC class I components or transcription factors that govern IFN responses. For instance, alternative splicing of *STAT1* generates distinct isoforms that activate unique gene expression programs downstream of IFNγ signaling [23, 24]. Elucidating these mechanisms in future studies will be crucial for fully understanding MBNL's role in anti-tumor immunity.

This emerging role of MBNL in tumor-immune interactions may present new therapeutic opportunities with broad clinical implications. MBNL proteins are known to drive the pathogenesis of myotonic dystrophy, where a trinucleotide repeat expansion sequesters MBNL, leading to widespread RNA splicing defects [17, 18,25–27]. Interestingly, two hallmark features of the disease—muscle wasting and cataract formation—have been linked to aberrant inflammatory activation [28, 29]. Combined with our findings, this suggests that MBNL1 could potentially regulate intrinsic immune responses in diverse cell types. Given that *MBNL* expression correlates with markers of T cell infiltration and activity in most human cancers, leveraging this relationship for therapeutic benefit could have far-reaching implications.

## Materials and methods

### Cell lines and culture

B16-F10 and E0771 cells were obtained from ATCC (CRL-6475 and CRL-3461) and MC38 cells were obtained from Sigma-Aldrich (SCC172). All cell lines were cultured per the manufacturer's instructions. Respective cell lines were infected at 0.3 MOI with Cas9 lentivirus (Addgene 52962-LV) and selected with 2 ug/mL Blasticidin for 7 days. Polyclonal cells were then single cell sorted, and individual clones were assessed for on-target editing efficiency, all subsequent CRISPR assays were completed with the respective Cas9 expressing clone with the highest editing rate.

To generate Cas9 edited mutant lines, we followed methods as previously described [30]. In brief, Cas9 expressing cells were transduced with a lentivirus containing a single guide RNA targeting either the respective gene of interest (Mbnl1, Mbnl2, etc.) or a paired-guide RNA targeting the two genes of interest (Mbnl1 and Mbnl2) in the pLentiguide-Puro backbone (Addgene #52963). Following transduction, cells with stable integration were selected using 1 ug/mL Puromycin for 72 hours.

### Animal use

All animal work and procedures were completed in accordance with the Guidelines for the Care and Use of Laboratory Animals and were approved by the Fred Hutchinson Cancer Center Institutional Animal Care and Use Committee (IACUC). Six week old male C57BL/6 mice were obtained from the Jackson Laboratory. In brief, animals for tumor studies were anesthetized using isoflurane gas until they were unresponsive for all initial tumor cell injections as approved by the

Fred Hutchinson Cancer Center IACUC. Animals were then routinely monitored throughout experiments by both core staff supported by Comparative Medicine and co-authors. When tumors reached the end point (1.5 cm in any dimension), the animals were sacrificed in accordance with institutional IACUC approved methods using CO2 inhalation for 5 minutes followed by cervical dislocation to attempt to minimize any suffering.

### *In vivo* validation studies

Cas9 expressing B16-F10 cells were grown per the manufacturer's recommendations and then transduced with lentivirus containing indicated pgRNAs and then selected in 1ug/mL Puromycin for 72 hours. $5\times10^6$ cells were then injected subcutaneously into each flank of adult male C57BL/6 mice, and monitored using calipers. Animals were euthanized when a tumor reached 1.5 cm in any dimension. Tumor material was isolated either for an archived flash frozen sample, fixed in 10% formalin at room temperature for histology studies, or placed in TRIzol reagent (Thermo Fisher 15596018) for subsequent RNA isolation.

### Immunohistochemistry

Tissues from tumors were processed, embedded, and stained through the Fred Hutch Experimental Histopathology core. Mouse Cd8a (CST Clone D4W27 1:200 dilution) staining was performed using rabbit monoclonal antibodies. Staining was performed with a BOND RX autostainer (Leica Biosystems) and images were acquired with an Aperio ImageScope at 40x magnification (Leica Biosystems). Image analysis was completed using HALO Image Analysis software.

### Western blotting

Total protein lysates were isolated in 1x RIPA buffer and quantified with the Pierce 660 nm Protein Assay Reagent (Thermo Fisher Scientific 22660). Total protein lysates were separated electrophoretically then transferred to a nitrocellulose membrane using the NuPAGE system (Thermo Fisher Scientific). Each membrane was blocked for 1 hour at room temperature and then probed with primary antibody diluted in a blocking buffer overnight at 4 degrees Celsius. Mbnl1 (DSHB Clone MB1(4A8)), Mbnl2 (Santa Cruz Biotechnologies Clone 3B4), and Gapdh (Abcam ab9485) primary antibodies were used. Anti-mouse or anti-rabbit IRDye (LI-COR Biosciences) secondary antibodies and the Odyssey CLx Imager (LI-COR Biosciences) were utilized for detection and imaging.

### In vitro flow cytometry and IFNγ stimulation

B16-F10 Cas9-expressing cells were then infected and grown for 48 hours. Cells were then plated and treated with standard DMEM with 10% FBS with indicated concentration of recombinant mouse IFNγ (Cell signaling technology 39127S). Following the indicated time course, all cells were washed 3x with PBS and then resuspended in 100 uL of PBS and stained with respective antibodies and LIVE/DEAD Fixable Violet (Thermo Fisher L34955) stain per the manufacturer's instructions. Cells were then passed through a 40-micron filter to generate a single cell suspension and then run on a BD FACSCelesta. Single cells were gated for live cells and then the signal was measured in the appropriate channel. Data were analyzed in FlowJo v10.

### In vivo immune cell and tumor cell flow cytometry

Tumors were dissected from mice and mechanically dissociated. Then 0.5 g of tissue were placed in 2-3 mL of tumor digestion buffer (20 mg/mL collagenase and 0.25 mg/mL DNAseI in 1x PBS) and shaken for 30 minutes at 37 degrees. The mixture was then subjected to centrifugation at 1200 rpm for 5 minutes, resuspended in 1x PBS, and then passed through a 70 micron filter (Corning 352250). Samples were then spun down and washed with 1x PBS and resuspended in 1x red blood cell lysis buffer (Thermofisher 00-4333-57). Cells were then washed with 1x PBS, these samples were then used

for flow cytometry of whole tumor cells. Cells were then incubated in 1x FACS buffer (1 mM EDTA and 2% FBS in 1x PBS) and 1:100 anti-H 2Kb/H-2Db antibody (BioLegend 114606).

For immune cell populations, digested and RBC lysed samples were then incubated with mouse CD45 microbeads (Miltenyi 130-052-301) and CD45+ cells were purified using positive selection columns per manufacturer's instructions (Miltenyi 130-042-401). Cells were then washed with PBS and then incubated with an antibody panel to identify T cell subsets and NK cells (CD90.2 BV786, CD8a eFluor 450, CD45 FITC, NK1.1 PE, CD4 BV510, anti-TCR-b chain Alexa Fluor 700) before proceeding with flow cytometry run on a BD FACSCelesta and analyzed in in FlowJo v10.

### gDNA PCR and on-target editing verification

gDNA was extracted with the DNeasy Blood and Tissue kit (Qiagen) according to the manufacturer's protocol. A window around the anticipated on-target cut site for the respective gene (Mbnl1, Mbnl2) was identified to allow for PCR amplification. PCR product was then purified and submitted for AmpliconEZ sequencing (Azenta/Genewiz). Reads were trimmed and mapped and the estimated fraction of reads with a specific indel were identified using the CRISPresso2 software using default settings [31].

### RNA-seq library preparation

Cells were first pelleted and then RNA was isolated using the Qiagen ReasyA kit (Qiagen 74104) and RNA purity was confirmed via 4200 TapeStation System. Poly(A)-selected, unstranded Illumina libraries were prepared according to the TruSeq protocol per manufacturer's instructions. The final library size and distribution was confirmed with a 4200 TapeStation System before sequencing on an Illumina HiSeq as 2x50 bp to generate ~40 million reads per sample.

### RNA-seq data analysis

RNA-seq was analyzed as previously described [32]. RNA-seq reads were mapped to an annotated transcriptome that was created using Ensembl 71 [33], UCSC knownGene [34] and Misov2.0 [35] annotations using RSEM version 1.2.4 [36] (modified to call Bowtie [37] with option '-v 2'). Unaligned reads were mapped to the corresponding genome (hg19/GRCh37 assembly, mm10/GRCmc38 assembly) as well as a database containing all possible pairings of 5' and 3' splice sites per gene contained in out merged transcriptome annotation using TopHat version 20.8b [38]. Lastly, mapped reads were merged and input into MISO v2.0. For TCGA studies, we analyzed 9,045 available samples across 29 cancer types.

### Cytolytic T cell correlations

RNA-seq data was analyzed as above and for each sample we quantified CD8A gene expression. For each individual cancer subtype we then correlated CD8A gene expression with expression of every coding gene in our lab annotation using Pearson correlation and generated a Pearson correlation coefficient. Then, for each gene we calculated the median Pearson correlation coefficient across all 29 analyzed cancer types.

### CRISPR screen re-analysis

CRISPR screen data was downloaded from previously published complementary screens in B16-F10 murine melanoma cells under distinct conditions to identify resistance and sensitizing mutations to antigen specific cytotoxic T cell killing [15].

### OT-1 cytotoxicity assay

Bulk splenocytes were isolated from OT-1 animals and cultured for three days in media containing 100 U/mL murine IL-2 and 100 μg/mL SIINFEKL peptide to activate CD8+ T cells. Cultures were then washed several times with media to

remove ova peptide and rested for at least 24 hours prior to use. OT-1 cells were passaged in T cell media with 50 U/mL IL-2 for no more than seven days after animal sacrifice before use. For the cytotoxicity assay, tumor cells alone or tumor cells + OT-1 cells (1:1 ratio) were incubated in T cell media for 18 hours under standard conditions and then analyzed by flow cytometry to quantify killing. OT-1 cells and other hematopoietic cells were excluded using CD45, CD3, and CD8 staining. Tumor cell viability was measured with DAPI staining.

## Supporting information

**S1 File. Supplementary figures and captions.**
(PDF)

**S1 Raw Images. Uncropped gel images.**
(PDF)

**S1 Table. Differential induction of IFNγ induced gene expression in NTC versus DKO B16-F10 Cas9 cells.** Differentially expressed genes from B16-F10 Cas9 cells infected with the indicated pgRNAs exposed to no or 1 ng/mL IFNγ for 24 hrs. Median gene expression per condition expressed in transcripts per million (TPM) along with the computed $\log_2$(fold-change) between the indicated comparison as well as the adjusted $P$ value.
(XLSX)

**S2 Table. Computed gene expression of MBNL proteins, CD8A, PRF1, and GZMB across TCGA RNA-seq samples.** Gene expression values (in transcripts per million, TPM) computed for all analyzed TCGA RNA-seq samples as well as the computed geometric mean of GZMB and PRF1 used as a proxy for cytolytic T cell activity.
(XLSX)

**S3 Table. Genome wide correlation of gene expression with CD8A expression across 29 cancer types.** Computed Pearson correlation coefficients and associated $P$ values comparing gene expression of the indicated gene with CD8A expression in the indicated cancer subtype. Each subtype is analyzed independently and the number of samples analyzed is documented per cancer subtype.
(XLSX)

## Acknowledgements

The results in this publication are based in part on data from The Cancer Genome Atlas Research Network (http://cancergenome.nih.gov). The funders had no role in study design, data collection and analysis, decision to publish, or preparation of the manuscript.

## Author contributions

**Conceptualization:** Austin M. Gabel, Omar Abdel-Wahab, James D. Thomas, Robert K. Bradley.

**Data curation:** Edie I. Crosse.

**Formal analysis:** Edie I. Crosse, James D. Thomas.

**Funding acquisition:** Robert K. Bradley.

**Investigation:** Austin M. Gabel, Andrea E. Belleville, Simon J. Hogg, Siegen A. McKellar, James D. Thomas.

**Visualization:** Edie I. Crosse.

**Writing – original draft:** Austin M. Gabel, Edie I. Crosse.

**Writing – review & editing:** Robert K. Bradley.

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
