## [Decision Letter · Decision Letter 0]

6 Jan 2025

PONE-D-24-15862Muscleblind-like proteins are novel modulators of the tumor-immune microenvironmentPLOS ONE

Dear Dr. Bradley,

Thank you for submitting your manuscript to PLOS ONE. After careful consideration, we feel that it has merit but does not fully meet PLOS ONE’s publication criteria as it currently stands. Therefore, we invite you to submit a revised version of the manuscript that addresses the points raised during the review process.

We look forward to receiving your revised manuscript.

Kind regards,

Atsushi Asakura, Ph.D

Academic Editor

PLOS ONE

Journal Requirements:

 “A.M.G and S.A.M are ARCS Foundation scholars. E.I.C is a Damon Runyon fellow. J.D.T. was supported by the NIH/NCI (K99 CA263168). R.K.B. was supported in part by the NIH/NCI (R01 CA251138), NIH/NHLBI (R01 HL128239, R01 HL151651) and the Blood Cancer Discoveries Grant program through the Leukemia & Lymphoma Society, Mark Foundation for Cancer Research, and Paul G. Allen Frontiers Group (8023-20). R.K.B is a Scholar of The Leukemia & Lymphoma Society (1344-18) and holds the McIlwain Family Endowed Chair in Data Science. Computational studies were supported in part by FHCC’s Scientific Computing Infrastructure (ORIP S10 OD028685). Experimental studies were supported in part by the Experimental Histopathology, Flow Cytometry, and Genomics Shared Resources of the Fred Hutch/University of Washington Cancer Consortium (NIH/NCI P30 CA015704). The results in this publication are based in part on data from The Cancer Genome Atlas Research Network (http://cancergenome.nih.gov).”

“A.M.G and S.A.M are ARCS Foundation scholars. E.I.C is a Damon Runyon fellow. J.D.T. was supported by the NIH/NCI (K99 CA263168). R.K.B. was supported in part by the NIH/NCI (R01 CA251138), NIH/NHLBI (R01 HL128239, R01 HL151651) and the Blood Cancer Discoveries Grant program through the Leukemia & Lymphoma Society, Mark Foundation for Cancer Research, and Paul G. Allen Frontiers Group (8023-20). R.K.B is a Scholar of The Leukemia & Lymphoma Society (1344-18) and holds the McIlwain Family Endowed Chair in Data Science. Computational studies were supported in part by FHCC’s Scientific Computing Infrastructure (ORIP S10 OD028685). Experimental studies were supported in part by the Experimental Histopathology, Flow Cytometry, and Genomics Shared Resources of the Fred Hutch/University of Washington Cancer Consortium (NIH/NCI P30 CA015704). The results in this publication are based in part on data from The Cancer Genome Atlas Research Network (http://cancergenome.nih.gov).”

“A.M.G and S.A.M are ARCS Foundation scholars. E.I.C is a Damon Runyon fellow. J.D.T. was supported by the NIH/NCI (K99 CA263168). R.K.B. was supported in part by the NIH/NCI (R01 CA251138), NIH/NHLBI (R01 HL128239, R01 HL151651) and the Blood Cancer Discoveries Grant program through the Leukemia & Lymphoma Society, Mark Foundation for Cancer Research, and Paul G. Allen Frontiers Group (8023-20). R.K.B is a Scholar of The Leukemia & Lymphoma Society (1344-18) and holds the McIlwain Family Endowed Chair in Data Science. Computational studies were supported in part by FHCC’s Scientific Computing Infrastructure (ORIP S10 OD028685). Experimental studies were supported in part by the Experimental Histopathology, Flow Cytometry, and Genomics Shared Resources of the Fred Hutch/University of Washington Cancer Consortium (NIH/NCI P30 CA015704). The results in this publication are based in part on data from The Cancer Genome Atlas Research Network (http://cancergenome.nih.gov).”

“RKB is a founder and scientific advisor of Codify Therapeutics and Synthesize Bio and holds equity in both companies. OA-W is a founder and scientific advisor of Codify Therapeutics and holds equity in this company. RKB and OA-W have received research funding from Codify Therapeutics unrelated to the current work. OA-W has served as a consultant for Foundation Medicine Inc., Merck, Prelude Therapeutics, Amphista Therapeutics, MagnetBio, and Janssen, and is on the Scientific Advisory Board of Envisagenics Inc., Harmonic Discovery Inc., and Pfizer Boulder; OA-W has received prior research funding from H3B Biomedicine, Nurix Therapeutics, Minovia Therapeutics, and LOXO Oncology unrelated to the current manuscript. The remaining authors declare no competing interests.”

7. We note that you have included the phrase “data not shown” in your manuscript. Unfortunately, this does not meet our data sharing requirements. PLOS does not permit references to inaccessible data. We require that authors provide all relevant data within the paper, Supporting Information files, or in an acceptable, public repository. Please add a citation to support this phrase or upload the data that corresponds with these findings to a stable repository (such as Figshare or Dryad) and provide and URLs, DOIs, or accession numbers that may be used to access these data. Or, if the data are not a core part of the research being presented in your study, we ask that you remove the phrase that refers to these data.

Reviewers' comments:

Reviewer's Responses to Questions

**Comments to the Author**

1. Is the manuscript technically sound, and do the data support the conclusions?

Reviewer #1: Yes

2. Has the statistical analysis been performed appropriately and rigorously? 

Reviewer #1: Yes

3. Have the authors made all data underlying the findings in their manuscript fully available?

Reviewer #1: Yes

4. Is the manuscript presented in an intelligible fashion and written in standard English?

Reviewer #1: Yes

5. Review Comments to the Author

Reviewer #1: I have carefully reviewed the manuscript submited to Plos One by Gabel AM and his colleagues. Their study entitled ‘Muscleblind-like proteins are novel modulators of the tumor- immune microenvironment’’ is that the effect of expression changes of a novel modulator, Muscleblind-like (MBNL) proteins, on the effect of CD8 T-cells on tumor cells and tumor- immune microenvironment was investigated.

The study is well designed, many experiments have been carried out within the framework of the hypothesis and very valuable data have been obtained from the experiments. These have been systematically given in the results section of the article, but it is seen that the presented data have not been discussed in detail in the discussion part of the article in the light of the literature. Therefore, the valuable data obtained from the study should be detailly discussed in the section of the article. Also, there are a few minor grammatical errors.

6. PLOS authors have the option to publish the peer review history of their article (what does this mean? ). If published, this will include your full peer review and any attached files.

**Do you want your identity to be public for this peer review?** For information about this choice, including consent withdrawal, please see our Privacy Policy .

Reviewer #1: No

---

## [Author Response · Author response to Decision Letter 1]

27 Feb 2025

Please see the response to reviewer and cover letter documents for point-by-point responses to reviewer and editorial requests.

---

## [Editor Report · Decision Letter 1]

3 Mar 2025

Muscleblind-like proteins are novel modulators of the tumor-immune microenvironment

PONE-D-24-15862R1

Dear Dr. Bradley,

We’re pleased to inform you that your manuscript has been judged scientifically suitable for publication and will be formally accepted for publication once it meets all outstanding technical requirements.

Kind regards,

Atsushi Asakura, Ph.D

Academic Editor

PLOS ONE
---

## [Editor Report · Acceptance letter]

PONE-D-24-15862R1

PLOS ONE

Dear Dr. Bradley,

I'm pleased to inform you that your manuscript has been deemed suitable for publication in PLOS ONE. Congratulations! Your manuscript is now being handed over to our production team.

Kind regards,

on behalf of

Dr. Atsushi Asakura

Academic Editor

PLOS ONE